# Relational responsibilities: Researchers perspective on current and progressive assessment criteria: A focus group study

Joeri K. Tijdink[1,2,3]*, Govert Valkenburg[4], Sarah de Rijcke[5], Guus Dix[6]

1 AmsterdamUMC, Department of Ethics, Health and Humanities, Amsterdam, the Netherlands, 2 Department of Philosophy, VU University, Amsterdam, the Netherlands, 3 Faculty of Psychology, Universitas Padjadjaram, Bandung, Indonesia, 4 Department of Interdisciplinary Studies of Culture, Norwegian University of Science and Technology, Trondheim, Norway, 5 Center for Science and Technology Studies (CWTS), Leiden University, Leiden, the Netherlands, 6 Faculty of Behavioural, Management and Social Sciences (BMS), Universiteit Twente, Enschede, the Netherlands

* j.tijdink@amsterdamumc.nl

## Abstract

### Introduction

The focus on quantitative indicators–number of publications and grants, journal impact factors, Hirsch-index–has become pervasive in research management, funding systems, and research and publication practices (SES). Accountability through performance measurement has become the gold standard to increase productivity and (cost-) efficiency in academia. Scientific careers are strongly shaped by the push to produce more in a veritable 'publish or perish' culture. To this end, we investigated the perspectives of biomedical researchers on responsible assessment criteria that foster responsible conduct of research.

### Methods

We performed a qualitative focus group study among 3 University medical centers in the Netherlands. In these centers, we performed 2 randomly selected groups of early career researchers (PhD and postdoc level & senior researchers (associate and full professors) from these 3 institutions and explored how relational responsibilities relate to responsible conduct of research and inquired how potential (formal) assessment criteria could correspond with these responsibilities.

### Results

In this study we highlighted what is considered responsible research among junior and senior researchers in the Netherlands and how this can be assessed in formal assessment criteria. The participants reflected on responsible research and highlighted several academic responsibilities (such as supervision, collaboration and teaching) that are often overlooked and that are considered a crucial prerequisite for responsible research. As these responsibilities pertain to intercollegiate relations, we henceforth refer to them as relational. After our systematic analysis of these relational responsibilities, participants suggested

**Data Availability Statement:** All relevant data are within the manuscript and its Supporting Information files.

**Funding:** ZonMw (= funder in the Netherlands) Award Number: Grant No. 445001010 | Recipient: Sarah de Rijcke, Prof. PhD The funders had no role in study design, data collection and analysis, decision to publish, or preparation of the manuscript.

**Competing interests:** The authors have declared that no competing interests exist.

some ideas to improve current assessment criteria. We focused on how these duties can be reflected in multidimensional, concrete and sustainable assessment criteria. Focus group participants emphasized the importance of assessing team science (both individual as collective), suggested the use of a narrative in researcher assessment and valued the use of 360 degrees assessment of researchers. Participants believed that these alternative assessments, centered on relational responsibilities, could help in fostering responsible research practices. However, participants stressed that unclarity about the new assessment criteria would only cause more publication stress and insecurity about evaluation of their performance.

## Conclusion

Our study suggests that relational responsibilities should ideally play a more prominent role in future assessment criteria as they correspond with and aspire the practice of responsible research. Our participants gave several suggestions how to make these skills quantifiable and assessable in future assessment criteria. However, the development of these criteria is still in its infancy, implementation can cause uncertainties among those assessed and consequently, future research should focus on how to make these criteria more tangible, concrete and applicable in daily practice to make them applicable to measure and assess responsible research practices in institutions.

## Trial registration

Open Science Framework https://osf.io/9tjda/.

## Introduction

The focus on quantitative indicators–number of publications and grants, journal impact factors, Hirsch-index–has become pervasive in research management, funding systems, and research and publication practices (SES). Accountability through performance measurement has become the gold standard to increase productivity and (cost-) efficiency in academia. Scientific careers are strongly shaped by the push to produce more in a veritable 'publish or perish' culture [1, 2]. Over the past decade, the shortcomings of assessment criteria for promotion and tenure for researchers have become increasingly evident [3–6]. Such criteria may stimulate undesirable research practices such as guest/ghost authorship, multi-publishing in predatory journals, salami slicing and p-hacking with the aim to increase performance metrics. These criteria jeopardize the responsible conduct of research. Besides, it may induce fierce competition, false incentives, reduced willingness to collaborate and reduced attention to other important academic duties (e.g. education). Moreover, other research also emphasized reform of evaluation practices to stimulate flexibility per individual assessment, institutions should incentivize non-traditional criteria [7] and emphasizing interdisciplinary work [8], would bring new perspectives to the debate on promotion and tenure. There have been a number of initiatives recently that aim to change current assessment practices (the Leiden Manifesto [6], the DORA statement [9], the changing incentives and reward criteria in funding bodies [10, 11], the Reward alliance and the Hong Kong manifesto [12].

However, two major unknowns still stand out.

First, it is unclear to what extent institutions will actually change their evaluation practices. A recent study, investigating such practices in 146 institutions, shows that, despite the criticism, many institutions still favor traditional assessment criteria over novel and innovative criteria that emphasize compliance with open science practices, good leadership, responsible supervision or educational skills [7]. Regarding the reliability of scientific knowledge, similarly, universities might remain rather conservative in not including the transparency, replicability and accuracy of reported findings in new-found assessment criteria [13].

Second, it is yet unclear whether the existing reform initiatives will counter unintended effects of the conventional use of performance metrics. Being relatively new, it is an open question whether or not they help to foster responsible conduct of research by selecting the scientists with more multidimensional profiles beyond a good publication, citation and funding record. Moreover, it is uncertain if these mechanisms will foster academic behaviour that complies with ideals of responsible conduct in research.

In light of these two unknowns, it is important to understand the kinds of bottom-up support that institutional changes in academic assessment can count on. For as a survey of publication practices showed, researchers do (re)consider such practices in relation to institutional evaluation criteria [14]. The dissemination and acceptability of new evaluation practices could benefit from insights into researchers' perspectives on the current assessment of responsible conduct of research and on the changes in the assessment of academic tasks and responsibilities that are needed to assess research and researchers responsibly. One study alluded to this by emphasizing researchers perceptions of valuation regimes beyond traditional research activities and how current systems and criteria can influence behavior and choices regarding the activities they prioritize [15]. This is also influenced by intense competition can narrow the range of activities and goals that scientists pursue due to the limited set of criteria and values by which scientists are judged and that reevaluating the criteria in academia is necessary to create a healthier and more sustainable research culture [16].

In a two-year qualitative research project, *Optimizing the responsible researcher*, we asked: What possibilities do biomedical researchers themselves see for the improvement of evaluation practices? To systematically address that question, we draw on 6 focus group interviews with junior and senior researchers at three Dutch University Medical Centers (UMC's).

## Methods

### Ethical approval

All procedures performed in this study involving human participants were in accordance with the ethical standards of the institutional and/or national research committee. The Medical Ethics Review Committee of VU University Medical Center assessed our project and confirmed that the Medical Research Involving Human Subjects Act (WMO) does not apply to our study (see Supplementary files).

### Participants

We included researchers who worked at 3 different University Medical Centers in the Netherlands. We decided to conduct focusgroups among 2 different groups and selected 2 types of active researchers; 1) Senior researchers: assistant, associate and full professors and 2) Junior researchers: postdocs, PhD candidates. We selected those researchers who have published about their research in the past two years, as a proxy criterion for their being 'active' as researchers. The two categories are considered sufficiently internally homogeneous with respect to academic seniority, and should allow for the focus group participants to recognize each other as peers. Our recruitment strategy was diverse; we used several techniques to invite

colleagues from different ranks and disciplines. We approached the dean's office to ask for interested researchers, used our collegial network and randomly invited researchers by email to invite them to participate in our study.

We recruited a variety of biomedical disciplines to create some perspectival diversity. Within each group, all participants were selected from the same UMC. We aimed for a group size between 6 and 8 participants, and maintained a minimum of 3 researchers. After confirmation of participation, the participants were sent a one-page description of the purposes of the research project as well as the main conditions of the informed consent in advance of the project by email.

## Informed consent

Participation to the focus groups were subject to obtaining verbal informed consent at the start of the focus group, referring to the sent document that contained the privacy policy and based on additional verbal explanation during the focus groups of the following conditions: 1) the purpose of the research and the focus group; 2) treatment of the focus group recordings: anonymized transcriptions and notes on nonverbal communication and behaviour, coding of transcriptions by the research team. And 3) the right to leave the focus group at any point, or to refuse answering specific questions or discussing specific topics, without further obligation of explanation.

## Procedure of focus groups

First, we investigated how researchers perceive the norms they see their work subject to, and how they see these norms relate to their own ideals of proper research / responsible research practices. Underlying the focus group methodology is the idea that people come to more balanced and comprehensive account in a conversation with peers, than they would in an individual conversation with an interviewer who is less akin. As set out above, the groups were composed homogeneously with respect to academic rank so participants could recognize each other as peers. Heterogeneity in biomedical disciplines was pursued, as this is reckoned conducive to critical appraisal.

The focus groups were conducted in 2017 and 2018. The focus groups were in English if there were participants that were not fluent in Dutch. A moderator facilitated the focus groups (JKT) and an observer (GV and/or GD) made notes about the process and its content. The focus groups started with a short introduction in which the goals of the project were explained. Questions could be answered in that phase. We used a topic guide as a semi-structured way to guide the focus group discussions.

The focus groups were recorded by the research team and transcribed by an external party. Most transcriptions were coded by 2 members of the research team (JKT and GD). In first instance, code systems were developed and via inductive content analysis led to several themes, and in second instance the themes were analysed by the second coder. After the focus groups, participants received a short summary of the discussion (member check) [17] or received a preprint of this manuscript.

## Analysis

We used inductive content analysis for the analysis of the transcripts of the focus groups and took the following steps in this process. First we coded the data on a phrase level. Next, we established a tentative set of themes to analyze the selected phrases, while allowing for flexibility to revise and update these themes during the process, if necessary. Then we analyzed the different themes and summarized these themes. This process of inductive content analysis

**Table 1. Demographic information of focus groups participants.**

| Academic rank | Junior Researchers | Senior researchers |
|---|---|---|
| **UMC** | | |
| UMC 1 | $4^{/3}$ | $4^{/3}$ |
| UMC 2 | $3^{/1}$ | $7^{/4}$ |
| UMC 3 | $6^{E/5}$ | $8^{E/4}$ |

$^{E}$ = focus group was conducted in English; all other focus groups were conducted in Dutch.

$^{/x}$ = number of female participants. UMC = University Medical Center

helps us to understand complex discussions and bring them back to meaningful themes [18]. One researcher (JKT) analyzed and coded the transcripts entirely. One other team member (GD) read the transcripts entirely and coded 2 transcripts independently. The themes were discussed with the other team members [17].

**Descriptive information.** We conducted 6 focus groups with 32 researchers from 3 university medical centers and across a wide variety of disciplinary fields, see Table 1.

## Results

We started the focus groups with an exercise to familiarize the participants with each other, the topic and the methodology. Participants were asked about their experiences with questionable research practices and what they find the most detrimental research misbehaviors. In both junior and senior groups, most participants had experiences with authorship disputes, while some of them reported more severe breaches of RI (such as data manipulation). Interestingly, bad supervision was mostly mentioned by the more senior participants as a potential threat of research integrity.

In analyzing the content of the focus group transcripts, we came to identify 3 themes–collaboration, supervision and teaching–that relate to responsible research practices. Each theme represents a responsibility of researchers vis-à-vis others that the participants think should be better recognized and rewarded. After this, we explored potential solutions for improving the assessment criteria that they are also in line with the identified themes of the responsible research practices.

Based on these 3 themes that were discussed, we divided them into 2 categories: 1) responsibilities of researchers that are related to research integrity; and 2) new forms of assessment that could cater to these responsibilities. Both categories hence consisted of 3 themes each. We have collected quotes from all focus groups and have reported the most typical quotes that illustrate the discussed solutions in Table 2 at the end of this results section. If the focus group was conducted in Dutch, we have translated the quote into English. Below, we present the two categories of themes. After these results of the focus groups, we will briefly discuss the results of the evaluation labs with relevant stakeholders that we organized to safeguard the uptake of our findings.

### I. Responsible research as the relational responsibilities of researchers

Research integrity is a term that sparked lively discussions in the focus groups. But it is also a term that was difficult to capture as one 'thing'; a thing that can be clearly defined and assessed. In the first part of the focus group, participants discussed what research integrity meant to them. They highlighted that a researcher with integrity is someone who does good research and knows what good research is. For many, research integrity is not necessarily a character

**Table 2. Overview of topics and quotes from the focusgroups conducted in our study.** Below we highlight the 3 main themes (collaboration, supervision and teaching by emphasizing the most suitable quotes about these themes. Furthermore, we describe the 3 solutions that were brought up during the discussions on researcher assessment in the FGs and highlight potential limitations.

| Responsibilities of researchers that are related to research integrity |
|---|
| ***Theme 1: Collaboration should be recognized in alternative assessments*** |

Senior Researcher UMC 1:
*And integrity: I cannot see yet how one should assess this. It is very important, of course, for a responsible scientist, to have some of these measures that you should ask for in a more qualitative manner or that should be part of the way people assess one another*

Senior Researcher UMC 2:
*The only thing that comes to my mind that would be an unbiased, but really tricky, way to sort of assess honesty, is to look at how someone's work is followed up and reproduced independently, right, after publication*

Junior researcher UMC 3:
*'being responsible for your researchers is also part of responsible research'*

Junior researcher UMC 2:
*'people who enjoy the game, but that are not doing responsible research, they stick around too much'*

Junior researcher UMC 1:
*'When you look at the system as it is right now, it presorts quite significantly on a very specific, scary personality type'*

Senior researcher UMC 3:
*'collaboration, the competence to collaborate, has to be assessed'*

senior researcher UMC 3:
*'You could also have like, well, a prize for a well-functioning team; you could have something like a team player prize'*

| ***Theme 2: PhD Supervision should be recognized in alternative assessments*** |

Senior researcher UMC 3:
*'well, you know how things ought to be and show that too. And you straighten people out when they think they can deviate from it'*

Senior researcher UMC 3:
*'not only have to invest in scientific training but in someone's personal development in the wider sense of the term'*

Senior researcher UMC 3:
*'you grow into this at a certain moment' through the 'interaction with your people'*

Senior researcher UMC 3:
*'when you are trained well and when it is, kind of, explained to you'*

| ***Theme 3: Teaching should be recognized in alternative assessments*** |

Junior researcher UMC 1:
*'To teach or to disseminate things to society: that does belong to it but is currently not considered to be a way to be a good scientist'*

Senior researcher UMC 3:
'For some people, teaching is not their greatest hobby. And then they still do it and try to somehow get a good evaluation out of it'

Senior researcher UMC 3:
'You have to conduct research, and create societal relevant work and you have to be good at teaching too'

| From Relational responsibilities to relational assessment |
|---|
| **Solutions** |
| Research culture, formation of teams and activities of team leaders |

| Quote: | Topic: |
|---|---|
| Senior researcher UMC 2:<br>*'In a research group, you function as a role model in that you also, well, know how things should go and show how things should go. And to call people to order in case they think they can deviate from this.'* | The important role of team leaders on research culture |
| Senior researcher UMC 2:<br>*'I would think that 'responsible' is more applicable to someone who intends to be of service to others and does not act in too selfish a way'* | The important role of team leaders on research culture |

*(Continued)*

**Table 2.** (Continued)

| | |
|---|---|
| *Junior Researchers UMC 1:* <br> *'narcissism is an outgoing personality, that is not something compatible with responsible researchers* | Relational responsibilities may not be compatible with certain personalities |
| Senior researcher UMC 3: <br> *'Because it is almost the case that you remain a team player despite the system. Because the system holds individuals accountable of course'* | The important role of formation of teams on research culture |
| ***Relational Assessment theme 1: Reward team effort and team science*** | |
| *Quote Senior Researcher UMC 2:* <br> *'According to me, the biggest bottleneck is that these criteria always apply to the individual. While we, in this conversation, emphasize that it is a collective effort, a team process'* | |
| ***Relational assessment theme 2: provide 360 degrees of feedback*** | |
| Quote Senior research UMC 1: <br> *'perhaps people should be interviewed by commissions or something like that. That you get a way better picture of the person and how he operates in such a team'* | |
| ***Relational assessment theme 3: engage in narrative assessment*** | |
| Quote Senior research UMC 1: <br> *'perhaps people should be interviewed by commissions or something like that. That you get a way better picture of the person and how he operates in such a team'* | |
| Senior Researcher UMC 2: <br> *'These soft skills are very difficult to capture in criteria that you can also put on a list'.* | |
| **Potential limitations of alternative assessments** | |
| *Quote* Senior researcher UMC 3: <br> *'shouldn't you be afraid, at least I am, that we are going to design a whole system to get the non-responsible people in line? That the instrument is in fact a bit of an overkill?'* | Alternative assessment can also be a bureaucratic burden |
| *Quote* Junior researcher UMC 1: <br> *'a little narcissism here and there is not so bad in itself because it will get you a long way and you can do many useful things with it in the end'* | Individual recognition is not solely bad but may also have positive consequences |

trait that you develop–although honesty would be part of the requirements of a responsible researcher. More than a character trait, integrity is seen by participants as an element of good research practices associated with the replicability of someone's work; the compliance with open science principles (such as sharing data); and collegial reviews to assess to what extent the published work is of high quality. Despite these elements, familiar ones for those professionally preoccupied with the topic, participants found it hard to make research integrity converge into a clear measure that could find its way in the assessment criteria. As a senior researcher (UMC 1) expressed it: *And integrity: I cannot see yet how one should assess this. It is very important, of course, for a responsible scientist, to have some of these measures that you should ask for in a more qualitative manner or that should be part of the way people assess one another.*

The reproducibility of scientific results, for instance, seems particularly hard to evaluate: '*The only thing that comes to my mind that would be an unbiased, but really tricky, way to sort of assess honesty, is to look at how someone's work is followed up and reproduced independently, right, after publication*' (Senior Researcher UMC 2).

But although research integrity seems elusive as a characteristic of research or of a researcher which one could directly measure and assess, participants did find indirect ways that open it up for evaluative purposes. What particularly stands out are the social relations and skills that are deemed necessary to act responsibly in research settings. These relations and skills are pivotal in conducting responsible research and permeate the kinds of role modeling and leadership that people are capable of. Or, in the words of a junior researcher, '*being*

*responsible for your researchers is also part of responsible research'* (Junior researcher UMC 3). Junior researchers would appreciate more focus on scientists who act responsibly because *'people who enjoy the game, but that are not doing responsible research, they stick around too much'* (Junior researcher UMC 2). Another junior researcher sides with this point of view, though expressed somewhat stronger terms: *'When you look at the system as it is right now, it presorts quite significantly on a very specific, scary personality type'* (Junior researcher UMC 1). A possible explanation for that presorting process, is that people who enjoy 'the game' sometimes become full professor predominantly on the basis of their skills as researcher, not so much on their skills as research leaders. Perhaps, these research content-related qualities should not stand out in the decision to let researchers 'supervise these people who end up at home with a burn-out' (Junior UMC 3). In addition to the inability to directly assess research integrity, the importance of social relations and skills in responsible research practices hence stood out as a second key finding. These relational responsibilities branch out in three different themes that should be (more) central in responsible research assessment according to our participants: researchers relate to peers in constructive collaboration to foster responsible research among researchers in biomedical research overall; researchers relate to PhD's in responsible supervisory practices; and they relate to students in responsible teaching practices.

**Relational responsibility theme 1: Collaboration.** The participants agreed that collaboration in general is essential for scientific research. In biomedicine, research is mostly done in teams and the importance and value 'team science' has gained more attention recently [19, 20]. Participants consider the ability to collaborate a pivotal skill for a biomedical researcher. They feel that poor collaborative skills are important reasons for disputes and potentially threaten research integrity. Having collaborative skills can, according to most of the participants, hence be considered as a core characteristic of a responsible researchers and therefore *'collaboration, the competence to collaborate, has to be assessed'* (senior researcher UMC 3). The actual assessments of (the development of) this competence could increase the emphasis on the collaborative nature of biomedical research.

The question how to integrate this competence in individual assessments leads to various answers. As with supervision, one can ask for 360 degrees of feedback from team members to assess someone's collaborative skills. Interestingly, some participants are geared towards more competitive inducements for collaboration: *'You could also have like, well, a prize for a well-functioning team; you could have something like a team player prize'* (senior researcher UMC 3). Other participants, on the contrary, reckoned that rewarding collaboration skills through such external incentives might be problematic due to the fact that collaboration should remain an intrinsic drive of team leaders: *'Well, many people have the inner incentive that they want their group to function well and we don't need to have such an artificial score for it'* (Senior Researcher UMC 1).

**Relational responsibility theme 2: PhD supervision.** In addition to collaboration, the participants stress that PhD supervision is one of the key elements of responsible research practices. Although it is not valued and represented in their institutions' assessment criteria, most participants feel that supervision is one of the cornerstones of such practices. To start with, they argue one has to be aware of the fact that one is a role model in their research group in the sense that *'well, you know how things ought to be and show that too. And you straighten people out when they think they can deviate from it'* (senior researcher UMC 3). In addition, participants consider that supervision of high quality should not solely focus on transmitting the technical knowledge and know-how that are essential for good research practices but should also focus on the relationship between supervisor and supervisee. They suggest that there should be attention to the personal development of the supervisee. As one senior researcher expressed it, to educate researchers to stand their ground in society you *'not only*

*have to invest in scientific training but in someone's personal development in the wider sense of the term'* (senior researcher UMC 3). Such supervision skills require both training and practice to make people aware of their role as well as their capacities. Being promoted from junior to senior positions *'you grow into this at a certain moment'* through the *'interaction with your people'* (senior researcher UMC 3). And even though personality may play a role here as well, becoming a good role model and developing supervisory skills is easier *'when you are trained well and when it is, kind of, explained to you'* (senior researcher UMC 3).

Assessment of supervision is complex: it is difficult to measure the quality of supervision and there are limitations to assess supervision quality by PhD students because they have a dependent relationship with their supervisor. A potential solution to these issues is also suggested. There can be a mentor system, where PI's from other departments have regular meetings with PhD students, or an semi-anonymous '360 degrees assessment' with special attention to supervision skills.

**Relational responsibility theme 3: Teaching.** Currently, educational responsibilities are often seen as a burden for individual researchers because they are not a top priority for them. Although the education of students is a vital responsibility of universities, this is not automatically reflected into the ideals of good science and responsible research practices: *'To teach or to disseminate things to society: that does belong to it but is currently not considered to be a way to be a good scientist'* (Junior researcher UMC 1). According to the participants, teaching has thus a very important indirect relation with responsible research practices. Students learn to become responsible researchers by following workshops and lectures, being supervised during master theses, develop academic skills in classrooms and gain experience in conducting research that is taught according to the highest standards of research integrity. However, engaging in education will cost researchers time and energy that they cannot spend on other 'assessable' tasks such as writing publications and grant applications. The notion that teaching will not be very important to their career advancement is widespread. This in turn makes that some participants have developed a pragmatic attitude towards teaching that can border on the opportunistic: *'For some people, teaching is not their greatest hobby. And then they still do it and try to somehow get a good evaluation out of it'* (senior researcher UMC 3). What does not help, participants add, is that hiring committees seem to be looking for the impossible. As one of our senior researchers states: *'You have to conduct research, and create societal relevant work and you have to be good at teaching too'* (Senior researcher UMC 3). But while you are expected to excel in all academic tasks, it is not easy to be good at everything.

The participants give several solutions to include education in the assessment system. This can be done by taking the hours spent on education, the development of new courses and student evaluations explicitly into consideration when assessing researchers. A possible limitation to these suggestions is also brought up: it may strengthen the pragmatic or opportunistic attitude to get high assessment scores on educational skills.

## From relational responsibilities to relational assessment

The three relational responsibilities that our participants singled out are deemed essential in fostering responsible research practices and a positive research culture of a faculty, department, or research group. However, the participants acknowledge that relational responsibilities very much depend on research culture, formation of teams and activities of team leaders. They are key in engaging in, and maintaining, collaborative relations, supervise young researchers and are often considered a role model to give the right example in their team. *'In a research group, you function as a role model in that you also, well, know how things should go and show how things should go. And to call people to order in case they think they can deviate from this.'*

(Senior researcher UMC 2). The orientation towards the team and how it functions is bound up with an orientation towards the kinds of social skills, such as a reflexive stance and communicative skills that someone has to possess or develop. Some participants highlighted that what makes that someone becomes a team player is a specific attitude. As way of relating to others, taking relational responsibility is opposed to the search for self-aggrandizement that is also plentiful in academia: *'I would think that 'responsible' is more applicable to someone who intends to be of service to others and does not act in too selfish a way'* (Senior researcher UMC 2). Or even more strongly put, responsible researchers can be opposed to narcistic attitudes of researchers: *'narcissism is an outgoing personality, that is not something compatible with responsible researchers.'* Though deemed essential to responsible research practices, team-oriented attitudes are not seen as core characteristics that are essential for researchers. If they want to further in their career, team oriented is something that is often being valued but never recognized in current evaluation forms: *'There are quite a lot of researchers who really make it, but who are not team players'* (Junior Researcher UMC 2*)*. On the contrary, the 'system' seems to discourage team science by focusing on individual recognition: *'Because it is almost the case that you remain a team player despite the system. Because the system holds individuals accountable of course'* (senior researcher UMC 3). Currently, the assessment criteria are predominantly focusing on achievements made by an individual. But when the research community deems relational responsibilities as important as individual achievement, then the assessment criteria should take such responsibilities into account. Below we describe the 3 themes that were highlighted by our participants as potential instruments to assess relational responsibilities.

**Relational assessment theme 1: Reward team effort and team science.** The current assessment criteria are mainly focused on individual publications and metrics performance and not rewarding team efforts. Since collaboration is considered one of the cornerstones of responsible biomedical research, more emphasis on team science in assessment and promotion criteria would help. This will also put less pressure on individual performance and reward a wider range of researchers for their contribution. Currently, the researchers who value collective performance as at least as important as individual achievement and may not focus solely at obtaining tenure or full professorship, do not feel recognized while they are very much involved in collaborative projects. As one researcher suggests: *'According to me, the biggest bottleneck is that these criteria always apply to the individual. While we, in this conversation, emphasize that it is a collective effort, a team process'* (senior research UMC 3). A second way of better assessing teamwork would be to reward academic leaders who show that you can lead a flourishing team on several levels. Only if you can fulfill these criteria, you will have a chance to get promoted: '*That you say something like: you only become a leader when you can really keep a team on track. And that is what you have to show. That is one of the competences that is assessed. And this means that you can manage this team, it means that you can see the broader scope of a research field. . .'* (senior researcher UMC 1)

A third and final suggestion would be to better reward a diverse set of roles in teams. This is not often part of current assessment systems but is a crucial factor in good team work. You need a diverse team with a wide variety of research qualities–and persons who possess such qualities–to make your research endeavor both responsible and successful. Rewarding team efforts here means that you have eyes for all the different perspectives and types work that are needed in a team and think of a way to make visible how an individual human being functions inside it: *'And to really give weight to this without that Hirsch-index type of thing. I don't know how to exactly do that. So: recognize the diversity inside the team and your role in this'* (senior researcher UMC 3).

**Relational assessment theme 2: Provide 360 degrees of feedback.** In the focus group discussions, participants also addressed the advantages and disadvantages of the '360 degrees feedback'-model. On the positive side, this is a form of assessment where you can explore multiple dimensions of a person such as collaborative skills, motivation and personality by (anonymously) interviewing co-workers, subordinates, or bosses. One participant highlighted one of the benefits: *'my supervisor did not see all my qualities. Other ranks should have a better picture of your qualities and what you can improve* (senior researcher UMC 3)'. A senior researcher, who is part of a committee that keeps track of PhD student well-being, comments upon the *'enormous number of dysfunctional labs, that you would never think of'* (Senior researcher UMC 2). To get a grip on existing lab culture and the problems that might otherwise remain hidden, the participant advocates 'the idea of interviewing also people who are junior' to get unbiased and open feedback: '*I mean, personally I also did this and it's really interesting to see what people think of me, what I need to tone down, what I need to improve on, and I think that's something that would be important in the assessment of senior PIs*' (senior researcher UMC 2). Introducing 360 degrees of feedback would allow the institute to find out what people think of the leadership, the environment and the (lack of) transparency.

On the downside, it can potentially be influenced by negative attitudes of others who are begrudged, are unkind and can be subjective. Another limitation is the fact that it is time consuming for both the institution and the researcher to complete. Among the participants, this sparked the discussion on audits, a thorough (and time-consuming) way to explore responsible research practices on the shop floor by auditing research teams and practices in laboratories. The participants directly stated that for some disciplinary fields this would be easier to implement than other fields (e.g. audits may potentially have most impact in research labs, and may be less feasible for departments that conduct clinical research).

**Relational assessment theme 3: Engage in narrative assessment.** A third way of assessing whether a researcher is responsible according to our participants is by using the narrative in assessment procedure. In narratives you assess the quality of a researcher in terms of research aims, individual motivation and responsible research practices and can answer the following questions. How does someone behave vis-à-vis others? What do colleagues say about the research or researcher? How compliant is their engagement with open science practices and what are the virtues of someone as a person? In line with what we remarked earlier, our participants were ambivalent about the idea that relational responsibilities can actually be directly assessed as they are not so easy to detect or measure. Or, as one senior researcher expressed it: *'These soft skills are very difficult to capture in criteria that you can also put on a list'* (Senior Researcher UMC 2). But assessment in a narrative form could help out here: *'perhaps people should be interviewed by commissions or something like that. That you get a way better picture of the person and how he operates in such a team'* (Senior research UMC 1). Another participant pointed to the possibilities of a narrative assessment in differentiating between qualities someone possesses. This comes back to the notion that you will not have a one size, fits all assessment and that most researchers have different qualities. Some researchers in specific disciplines do not publish much but have other important academic qualities that should be acknowledged and considered. To get a hold on these qualities, a narrative can shine light on them. Only with this, you will get a better idea what the researcher is like and what his qualities are.

Although relational responsibilities are valued as part of responsible research behavior and practices, some ambivalence remains here. Some participants also raised concerns that these alternative assessments would lead to too much assessment procedures for researchers: *'shouldn't you be afraid, at least I am, that we are going to design a whole system to get the non-responsible people in line? That the instrument is in fact a bit of an overkill*?' (senior researcher

UMC 3). Other participants do not see the strife for recognition from others as an evil per se. They argue that *'a little narcissism here and there is not so bad in itself because it will get you a long way and you can do many useful things with it in the end'* (junior researcher UMC 1). This point of view is in a certain way partially incongruent with other findings as this does emphasize the need for individual recognition and fewer relational responsibilities from narcissists.

## Discussion

In general, current evaluation criteria in academia have received severe criticism in the past years [4–7, 9, 13]. Several initiatives have pleaded for a different approach when assessing research and researchers and often conventional metrics, such as the impact factor or the H-index are considered weak and unidimensional. However, the use of these metrics is widespread and engrained in academic culture [5]. In this study we highlighted what is considered responsible research among junior and senior researchers in the Netherlands and how this can be assessed in formal assessment criteria. The participants of the focus groups reflected on responsible research and highlighted several academic responsibilities (such as supervision, collaboration and teaching) that are often overlooked and that are considered a crucial prerequisite for responsible research. After our systematic analysis of these 'relational responsibilities', we focused on the solutions that the participants came up with: how can these duties be reflected in multidimensional, concrete and sustainable assessment criteria? Participants emphasized the importance of assessing team science, suggested the use of a narrative in researcher assessment and valued the use of 360 degrees assessment of researchers. Participants believed that these alternative assessments, centered on relational responsibilities, could help in fostering responsible research practices.

Our study sheds a new light on the assessment discussions as it is emphasizing the assessment of relational responsibilities. Besides, our study also provides a number of suggestions how these skills should be assessed and how these skills are closely related and connected to responsible research. Our findings also fit well with the national initiative in the Netherlands entitled *'room for everyone's talent'* that is initiated by the Dutch Royal Academy of Science and recognized by all universities, funding agencies and other stakeholders in the Netherlands. In their position paper, they plead for a radical change in assessing researchers with an emphasis on diversity, focus on quality, stimulating open science and encouraging academic leadership [21]. Although not specifically focusing on relational responsibilities as a different entity, the way these responsibilities should be recognized is reflected in their position paper.

### Interpretation of data

Our study identifies the need for alternative assessment criteria that are related to responsible research practices and outlines how relational responsibilities should be valued in the assessment of research and researchers. Our study proposes alternatives that, although they may be a bit more difficult to implement and requires that assessment procedures are revised so as to enable them to incorporate qualitative evaluations. This is possibly challenging in its novelty, and requires additional research to further develop these evaluations. Moreover, the suggestions from the participants can help policy makers to redesign their current assessment criteria. They also are a plea for more elaborate discussions on the criteria and invite policymakers to use a different strategy to assess researchers. For example, 360 degree evaluations can form the basis of the yearly evaluation conversations where researchers have to reflect on the responsibilities they have towards others–not just their achievements. Before we turn to the introduction of these new forms of assessment in institutional settings it is necessary to emphasize the limitations of these assessment themes. There are (at least) three.

First, our results are still a first step and these types of assessment criteria are still in its infancy and need rigorous testing before they can be put into practice. Furthermore, the participants recommend a blended approach in which both the conventional criteria (JIF, H-index, funding acquisition) as the new criteria should inform assessment of researchers. We thus should exercise caution to avoid discarding valuable elements or aspects when making comprehensive changes or judgments, as prematurely eliminating the essential along with the dispensable may lead to unintended consequences.

Second, taking disciplinary differences into account, we are aware that we only included biomedical researchers. However, we think that the emphasized relational responsibilities such as leadership, collaboration or supervision are not skills that are solely displayed in biomedical research and most likely the results can be generalized to other disciplines.

Third, it is good to reflect on both reliability and validity of our findings in other research contexts (both disciplinary fields and other countries) since we only included 3 Dutch UMCs in our analysis. Future research could address these limitations by including other disciplines in our methods when designing and testing novel assessment criteria, systematically ask for reasons for non-response and try to organize larger focus groups in other disciplines.

Finally, the results also underline the relation between relational responsibilities and responsible research. Earlier research concluded that responsible research is strongly shaped by research cultures. In these cultures, norms and unwritten rules play a significant role in how researchers behave and which social skills are encouraged [21–24]. Assessing these skills may have an impact on more responsible research practices on the shop floor.

We also would like to highlight the CRediT taxonomy initiative as an initiative to improve researcher assessment. The CrediT (Contributor Roles Taxonomy) is a classification system developed to provide standardized categories for describing the contributions of individuals to scholarly work. It was created to address the need for more transparent and granular acknowledgment of individual contributions within scholarly publications. The taxonomy outlines various roles that individuals may play in the research process, going beyond traditional authorship credits. This is a better way to give credit where credit is due and further details on the roles authors play and thus give better insight in roles that can help in the assessment of researchers. Another welcome initiative is CoARA, the Coalition for advancing Research Assessment. This European coalition tries to change the current assessment system by improving the assessment of research, researchers and research organizations by recognizing and rewarding the diverse outputs, practices and activities that maximize the quality and impact of research. This requires basing assessment primarily on qualitative judgement, for which peer review is central, supported by responsible use of quantitative indicators.

## Evaluation labs

Since our project only explored the perspective of biomedical researchers, we wanted to corroborate whether these newly formulated evaluation ideas actually can be implemented in real life settings. Therefore, we conducted two evaluation labs. In these labs, we invited stakeholders (policy makers working at funding bodies in the Netherlands, and policy makers working at the eight University medical centers) to discuss the results of our study and let them reflect on whether the suggestions are implementable and feasible. They were encouraged to think what solutions that were highlighted by the participants of the focus groups could actually be useful for their institution. We asked them to discuss whether and how the themes from the focus group study could help shape new assessment criteria in their institution (exercise 1). After the discussion of the themes, we asked them to rank the themes on importance and feasibility (exercise 2).

**Exercise 1.** The participants think that integrity should be an overall assessment criterium that should be included in all themes and included the work environment as part of research integrity. They agree that assessing *relational responsibilities* remains a very complicated matter as the participants consider assessing them as something subjective and incomplete. However, they do consider it potentially interesting to assess leadership skills as relational responsibility. They specifically highlighted supervision, and the participants suggested that PhD students should evaluate the supervisor and this knowledge should be available to include in promotion and assessment.

As for assessing relational responsibilities, they suggest that *360 degree assessment* can be an interesting tool and should be expanded to international colleagues as well. Furthermore, they suggest to include the evaluation of the researchers by their patients and students as well. Finally, the assessment by *narrative* has the potential to be important. The main concern is that it may expand the workload of researchers even more with administrative work. It is also prone for opportunistic behaviour and manipulation. Besides, achieving trust in the narrative-method can be challenging.

**Exercise 2.** We asked the participants to rank the themes by whether they consider this the most essential theme. After careful deliberation, the participants ranked collaboration, integrity and supervision as most essential. After this exercise, they included diversity as an important theme that is extensively discussed in the past years and should be considered as an evaluation criterion. Research groups can benefit from diversity with respect to competences and qualities.

Two main concerns remain in place according to the participants. In their discussions they repeatedly highlighted that there is a natural overlap between the teams and some themes are also covered by other themes (e.g. Is good supervision part of research integrity?). They also noted that scientific rigor and quality are not assessed anywhere and suggested that this should be operationalized as they relate closely with responsible research and are connected with the research integrity theme.

## Strengths and limitations

There are some strengths and limitations to this study to bear in mind. First of all, this is the first study that investigates responsible research as a criterion in assessment of researchers in a qualitative way. Qualitative methods are helpful if you want to explore new perspectives or formulate new hypothesis and offers a complimentary contribution to prior quantitative and hypothesis driven research. Secondly, the participants highlight that it is possible to assess relational responsibilities. Since the current practices predominantly rely on quantitative output criteria, this is an important message that may help policymakers to adopt alternative criteria in their assessment procedures. Finally, we were able to test the results of the focus groups in the evaluation labs. This has resulted in interesting thoughts on potential implementation issues, but shine also new light on the discussion of what is actually possible, warranted and needed for policy makers and funders.

Our study also comes with some limitations. First, we only included biomedical researchers. Although we are convinced that most criteria can be generalized to other disciplinary fields, one may argue that in other disciplines (e.g. the humanities) slightly different assessment criteria might apply. Biomedicine was purposefully considered as a strategic research site. Evidence suggests that biomedical research assessment is prone for quantification [25]. Also, biomedicine is the largest discipline in the Netherlands (and abroad). Besides, dominant organisational forms in biomedical research—research projects, groups and labs—are increasingly found in the social sciences and humanities as well [26, 27]. Our findings will thus have implications

beyond biomedicine. Future research is thus needed to explore these potential disciplinary differences and should test the proposed assessment criteria. Second, a form of response bias cannot be ruled out. We have sent out invitations to a large set of participants. Most of the invited researchers were asked for the reason for non-participation. Most often, lack of time was the main reason not to participate. Third, it could be that we are still missing perspectives. Some focus groups were rather small and although we reached data saturation for most themes, it could well be that some perspectives are not highlighted.

## Changing the system

It is a daunting task to change assessment criteria that have been used for decades in various institutions. The academic enterprise is a complex system and change is often met with a certain skepticism, both by established researchers that have made careers by older assessment criteria (survivor bias) and by the fear that future assessment criteria will change academia into a system where only 'soft skills' are assessed. However, change starts with awareness and it helps that there are successful initiatives that have shown their value. Besides, implementation has started by some of the most important stakeholders. From our results we draw the conclusion that responsible research, constituted of both relational responsibilities and more quantifiable assessment criteria, should be a major element of researchers assessment in the future. This entails both 1) rigorous testing of reformed assessment criteria before implementing them in practice and 2) communicating this transparently with researchers, so they know what to expect in future assessments. More attention is still needed on relational responsibilities that, taken into account that they are highly influencing responsible research, should therefore play a more prominent role in future assessment criteria.

In conclusion, our study suggests that relational responsibilities should ideally play a more prominent role in future assessment criteria as these relational responsibilities are reckoned beneficial for the practice of responsible research. Our participants gave several suggestions on how to make these skills measurable and assessable in future assessment criteria. However, the development of these criteria is still in its infancy and future research should focus on how to make these criteria more tangible, concrete and applicable in daily practice.

## Supporting information

**S1 File. Medical ethical review Approval_METC_Tijdink.**
(PDF)

**S1 Table. Translation Dutch quotes focus groups.**
(DOCX)

## Author Contributions

**Conceptualization:** Joeri K. Tijdink, Govert Valkenburg, Sarah de Rijcke.

**Data curation:** Joeri K. Tijdink.

**Formal analysis:** Joeri K. Tijdink, Govert Valkenburg, Guus Dix.

**Funding acquisition:** Joeri K. Tijdink, Govert Valkenburg, Sarah de Rijcke.

**Investigation:** Joeri K. Tijdink, Sarah de Rijcke, Guus Dix.

**Methodology:** Joeri K. Tijdink, Govert Valkenburg, Guus Dix.

**Project administration:** Sarah de Rijcke.

**Software:** Joeri K. Tijdink.

**Supervision:** Govert Valkenburg, Sarah de Rijcke, Guus Dix.

**Validation:** Joeri K. Tijdink, Guus Dix.

**Visualization:** Joeri K. Tijdink, Guus Dix.

**Writing – original draft:** Joeri K. Tijdink, Guus Dix.

**Writing – review & editing:** Joeri K. Tijdink, Govert Valkenburg, Sarah de Rijcke, Guus Dix.

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
