## [Editor Report · Decision Letter 0]

6 Sep 2023

PONE-D-23-25545Title: Relational responsibilities: Researchers perspective on current and progressive assessment criteria: a focus group studyPLOS ONE

Dear Dr. Tijdink,

Thank you for submitting your manuscript to PLOS ONE. After careful consideration, we feel that it has merit but does not fully meet PLOS ONE’s publication criteria as it currently stands. Therefore, we invite you to submit a revised version of the manuscript that addresses the points raised during the review process.

Dear author/s, thanks for submitting your work to PLoS ONE
,  I contrasted the core sections of your work with our seven criteria for publication and I consider you could work on the following points before sending it for review:  The study presents the results of original research Consider enrich the introduction/review with additional literature/evidence on the topic. This, in order to grownd the results and their discussion. For instance, authors could assess and potentially include the following studies to their literature: Fochler, M., Felt, U., & Müller, R. (2016). Unsustainable growth, hyper-competition, and worth in life science research: Narrowing evaluative repertoires in doctoral and postdoctoral scientists’ work and lives. Minerva, 54(2), 175–200. https://doi.org/10.1007/s11024-016-9292-y Robinson-Garcia, N., Costas, R., Nane, G. F., & van Leeuwen, T. N. (2023). Valuation regimes in academia: Researchers’ attitudes towards their diversity of activities and academic performance. Research Evaluation. https://doi.org/10.1093/RESEVAL/RVAC049  Experiments, statistics, and other analyses are performed to a high technical standard and are described in sufficient detail Consider expand the details on sub-section “Analysis.” The Technique of “Inductive content analysis” just mentioned, however, if it is the main technique to analyze the transcripts, at least, I assume, there are general steps to follow and explain (e.g., 1. Decide the level of analysis: word, word sense, phrase, sentence, themes; 2. Decide how various concepts to code for: develop a pre-defined or interactive set of categories or concepts. Decide either: A. to allow flexibility to add categories through the coding process, or B. to stick with the pre-defined set of categories; 3. Decide whether to code for existence or frequency of a concept. The decision changes the coding process; 4. Decide on how you will distinguish among concepts; 5. Develop rules for coding your texts; 6. Decide what to do with irrelevant information: should this be ignored (e.g. common English words like “the” and “and”), or used to reexamine the coding scheme in the case that it would add to the outcome of coding?; 7. Code the text [https://www.publichealth.columbia.edu/research/population-health-methods/content-analysis]). The whole section on the description and analysis of Table 1, should be part of the “Method” section, not “Results.”  The article adheres to appropriate reporting guidelines and community standards for data availability Consider expand on the reliability and validity of the methods and results got by using content analysis (https://www.publichealth.columbia.edu/research/population-health-methods/content-analysis): -
Reliability (Stability: the tendency for coders to consistently re-code the same data in the same way over a period of time; Reproducibility: tendency for a group of coders to classify categories membership in the same way; Accuracy: extent to which the classification of text corresponds to a standard or norm statistically), and -
Validity (Closeness of categories: this can be achieved by utilizing multiple classifiers to arrive at an agreed upon definition of each specific category. Using multiple classifiers, a concept category that may be an explicit variable can be broadened to include synonyms or implicit variables; Conclusions: What level of implication is allowable? Do conclusions correctly follow the data? Are results explainable by other phenomena? This becomes especially problematic when using computer software for analysis and distinguishing between synonyms. For example, the word “mine,” variously denotes a personal pronoun, an explosive device, and a deep hole in the ground from which ore is extracted. Software can obtain an accurate count of that word’s occurrence and frequency, but not be able to produce an accurate accounting of the meaning inherent in each particular usage. This problem could throw off one’s results and make any conclusion invalid; Generalizability of the results to a theory: dependent on the clear definitions of concept categories, how they are determined and how reliable they are at measuring the idea one is seeking to measure. Generalizability parallels reliability as much of it depends on the three criteria for reliability.) The article is presented in an intelligible fashion and is written in standard English Some readers might not understand colloquialism used in the manuscript, such as “We thus should not throw the baby out with the bath water.” Try to avoid them.  I hope you can incorporate the above suggestions to improve your already valuable work before sending it to review.  Sincerely,  Julián D. Cortés Associate Editor==============================

We look forward to receiving your revised manuscript.

Kind regards,

Julian D. Cortes

Academic Editor

PLOS ONE

Journal Requirements:

5. Please include a copy of Table 3 which you refer to in your text on page 8.

Additional Editor Comments:

Dear author/s, thanks for submitting your work to PLoS ONE ,

I contrasted the core sections of your work with our seven criteria for publication and I consider you could work on the following points before sending it for review:

The study presents the results of original research

Consider enrich the introduction/review with additional literature/evidence on the topic. This, in order to grownd the results and their discussion. For instance, authors could assess and potentially include the following studies to their literature:

Fochler, M., Felt, U., & Müller, R. (2016). Unsustainable growth, hyper-competition, and worth in life science research: Narrowing evaluative repertoires in doctoral and postdoctoral scientists’ work and lives. Minerva, 54(2), 175–200. https://doi.org/10.1007/s11024-016-9292-y

Robinson-Garcia, N., Costas, R., Nane, G. F., & van Leeuwen, T. N. (2023). Valuation regimes in academia: Researchers’ attitudes towards their diversity of activities and academic performance. Research Evaluation. https://doi.org/10.1093/RESEVAL/RVAC049

Experiments, statistics, and other analyses are performed to a high technical standard and are described in sufficient detail

Consider expand the details on sub-section “Analysis.” The Technique of “Inductive content analysis” just mentioned, however, if it is the main technique to analyze the transcripts, at least, I assume, there are general steps to follow and explain (e.g., 1. Decide the level of analysis: word, word sense, phrase, sentence, themes; 2. Decide how various concepts to code for: develop a pre-defined or interactive set of categories or concepts. Decide either: A. to allow flexibility to add categories through the coding process, or B. to stick with the pre-defined set of categories; 3. Decide whether to code for existence or frequency of a concept. The decision changes the coding process; 4. Decide on how you will distinguish among concepts; 5. Develop rules for coding your texts; 6. Decide what to do with irrelevant information: should this be ignored (e.g. common English words like “the” and “and”), or used to reexamine the coding scheme in the case that it would add to the outcome of coding?; 7. Code the text [https://www.publichealth.columbia.edu/research/population-health-methods/content-analysis]).

The whole section on the description and analysis of Table 1, should be part of the “Method” section, not “Results.”

The article adheres to appropriate reporting guidelines and community standards for data availability

Consider expand on the reliability and validity of the methods and results got by using content analysis (https://www.publichealth.columbia.edu/research/population-health-methods/content-analysis):

- Reliability (Stability: the tendency for coders to consistently re-code the same data in the same way over a period of time; Reproducibility: tendency for a group of coders to classify categories membership in the same way; Accuracy: extent to which the classification of text corresponds to a standard or norm statistically), and

- Validity (Closeness of categories: this can be achieved by utilizing multiple classifiers to arrive at an agreed upon definition of each specific category. Using multiple classifiers, a concept category that may be an explicit variable can be broadened to include synonyms or implicit variables; Conclusions: What level of implication is allowable? Do conclusions correctly follow the data? Are results explainable by other phenomena? This becomes especially problematic when using computer software for analysis and distinguishing between synonyms. For example, the word “mine,” variously denotes a personal pronoun, an explosive device, and a deep hole in the ground from which ore is extracted. Software can obtain an accurate count of that word’s occurrence and frequency, but not be able to produce an accurate accounting of the meaning inherent in each particular usage. This problem could throw off one’s results and make any conclusion invalid; Generalizability of the results to a theory: dependent on the clear definitions of concept categories, how they are determined and how reliable they are at measuring the idea one is seeking to measure. Generalizability parallels reliability as much of it depends on the three criteria for reliability.)

The article is presented in an intelligible fashion and is written in standard English

Some readers might not understand colloquialism used in the manuscript, such as “We thus should not throw the baby out with the bath water.” Try to avoid them.

I hope you can incorporate the above suggestions to improve your already valuable work before sending it to review.

Sincerely,

Julián D. Cortés

Associate Editor

---

## [Author Response · Author response to Decision Letter 0]

22 Nov 2023

Dear Prof. Cortés,

Thank you for your review of our article, recently submitted to PLoS One. We have read it with interest and below you will find answers in italics to the points you have raised in a point by point fashion. If we have changed the text, we indicate this and write this text in blue. This letter is also included in the attached files.

Thanks again for your valuable feedback. We look forward to hear from you.

Warm regards, also on behalf of the authors.

Joeri Tijdink, MD PhD

Amsterdam

Editor: I contrasted the core sections of your work with our seven criteria for publication and I consider you could work on the following points before sending it for review: 

1. The study presents the results of original research

Consider enrich the introduction/review with additional literature/evidence on the topic. This, in order to grownd the results and their discussion. For instance, authors could assess and potentially include the following studies to their literature: 

Fochler, M., Felt, U., & Müller, R. (2016). Unsustainable growth, hyper-competition, and worth in life science research: Narrowing evaluative repertoires in doctoral and postdoctoral scientists’ work and lives. Minerva, 54(2), 175–200. https://doi.org/10.1007/s11024-016-9292-y

Robinson-Garcia, N., Costas, R., Nane, G. F., & van Leeuwen, T. N. (2023). Valuation regimes in academia: Researchers’ attitudes towards their diversity of activities and academic performance. Research Evaluation. https://doi.org/10.1093/RESEVAL/RVAC049

Answer: Thank you for your thoughtful suggestion. We are happy you brought up these references as they have inspired us in the debate on evaluation and academic performance. We have expanded the introduction section with the two references and have added the following text to the introduction. We now write:

“One study alluded to this by emphasizing researchers perceptions of valuation regimes beyond traditional research activities and how current systems and criteria can influence behavior and choices regarding the activities they prioritize (14,15). This is also influenced by intense competition can narrow the range of activities and goals that scientists pursue due to the limited set of criteria and values by which scientists are judged and that reevaluating the criteria in academia is necessary to create a healthier and more sustainable research culture (15)”. 

2. Experiments, statistics, and other analyses are performed to a high technical standard and are described in sufficient detail

Consider expand the details on sub-section “Analysis.” The Technique of “Inductive content analysis” just mentioned, however, if it is the main technique to analyze the transcripts, at least, I assume, there are general steps to follow and explain (e.g., 1. Decide the level of analysis: word, word sense, phrase, sentence, themes; 2. Decide how various concepts to code for: develop a pre-defined or interactive set of categories or concepts. Decide either: A. to allow flexibility to add categories through the coding process, or B. to stick with the pre-defined set of categories; 3. Decide whether to code for existence or frequency of a concept. The decision changes the coding process; 4. Decide on how you will distinguish among concepts; 5. Develop rules for coding your texts; 6. Decide what to do with irrelevant information: should this be ignored (e.g. common English words like “the” and “and”), or used to reexamine the coding scheme in the case that it would add to the outcome of coding?; 7. Code the text [https://www.publichealth.columbia.edu/research/population-health-methods/content-analysis]). 

The whole section on the description and analysis of Table 1, should be part of the “Method” section, not “Results.” 

Answer: Thanks for this. We have extended our description of the inductive content analysis in the manuscript. Furthermore, we have moved the ‘descriptive information’ including table 1 to the methods section. We now write in the methods section under analysis on page 7:

“We used inductive content analysis for the analysis of the transcripts of the focus groups and took the following steps in this process. First we coded the data on a phrase level. Next, we established a tentative set of themes to analyze the selected phrases, while allowing for flexibility to revise and update these themes during the process, if necessary. Then we analyzed the different themes and summarized these themes. This process of inductive content analysis helps us to understand complex discussions and bring them back to meaningful themes (16)”.

3. The article adheres to appropriate reporting guidelines and community standards for data availability

Consider expand on the reliability and validity of the methods and results got by using content analysis (https://www.publichealth.columbia.edu/research/population-health-methods/content-analysis): 

- Reliability (Stability: the tendency for coders to consistently re-code the same data in the same way over a period of time; Reproducibility: tendency for a group of coders to classify categories membership in the same way; Accuracy: extent to which the classification of text corresponds to a standard or norm statistically), and 

- Validity (Closeness of categories: this can be achieved by utilizing multiple classifiers to arrive at an agreed upon definition of each specific category. Using multiple classifiers, a concept category that may be an explicit variable can be broadened to include synonyms or implicit variables; Conclusions: What level of implication is allowable? Do conclusions correctly follow the data? Are results explainable by other phenomena? This becomes especially problematic when using computer software for analysis and distinguishing between synonyms. For example, the word “mine,” variously denotes a personal pronoun, an explosive device, and a deep hole in the ground from which ore is extracted. Software can obtain an accurate count of that word’s occurrence and frequency, but not be able to produce an accurate accounting of the meaning inherent in each particular usage. This problem could throw off one’s results and make any conclusion invalid; Generalizability of the results to a theory: dependent on the clear definitions of concept categories, how they are determined and how reliable they are at measuring the idea one is seeking to measure. Generalizability parallels reliability as much of it depends on the three criteria for reliability.)

Answers: Thank you. Personally, I like this a lot to include this in an article as it tells us more about the quality and reproducibility of our work. We have added these considerations to the discussion section in the interpretation of the data. We have added the following text:

“Third, it is good to reflect on both reliability and validity of our findings. Considering the reproducibility of our work, two persons from our research team coded and analyzed the transcripts to assure that we found and classified similar categories and themes. Considering the validity, JT and GD have discussed the content of the themes in more detail on several occasions to assure that our categorization is similar and explicit. Moreover, we have discussed generalizability and reflect on this in the limitation section of this article.”

4. The article is presented in an intelligible fashion and is written in standard English

Some readers might not understand colloquialism used in the manuscript, such as “We thus should not throw the baby out with the bath water.” Try to avoid them. 

Answer: Thank you. We have changed this sentence and we now write in the discussion section: 

“We thus should exercise caution to avoid discarding valuable elements or aspects when making comprehensive changes or judgments, as prematurely eliminating the essential along with the dispensable may lead to unintended consequences.”

---

## [Decision Letter · Decision Letter 1]

13 Feb 2024

PONE-D-23-25545R1Title: Relational responsibilities: Researchers perspective on current and progressive assessment criteria: a focus group studyPLOS ONE

Dear Dr. Tijdink,

Thank you for submitting your manuscript to PLOS ONE. After careful consideration, we feel that it has merit but does not fully meet PLOS ONE’s publication criteria as it currently stands. Therefore, we invite you to submit a revised version of the manuscript that addresses the points raised during the review process.

Dear authors,

In view of the referees’ feedback and my own reading of your paper, we invite you to address all issues noted below, most of which are relatively minor in nature, but nonetheless essential. In particular, a few adjustments in literature review; counts/quantification of the focus group themes; and more discussion rooted in existing reward model structure.

We look forward to your revised version.

Sincerely,

Julián D. Cortés==============================

We look forward to receiving your revised manuscript.

Kind regards,

Julian D. Cortes

Academic Editor

PLOS ONE

Journal Requirements:

Reviewers' comments:

Reviewer's Responses to Questions

**Comments to the Author**

1. If the authors have adequately addressed your comments raised in a previous round of review and you feel that this manuscript is now acceptable for publication, you may indicate that here to bypass the “Comments to the Author” section, enter your conflict of interest statement in the “Confidential to Editor” section, and submit your "Accept" recommendation.

Reviewer #1: All comments have been addressed

Reviewer #2: (No Response)

2. Is the manuscript technically sound, and do the data support the conclusions?

Reviewer #1: Yes

Reviewer #2: Yes

3. Has the statistical analysis been performed appropriately and rigorously? 

Reviewer #1: Yes

Reviewer #2: N/A

4. Have the authors made all data underlying the findings in their manuscript fully available?

Reviewer #1: Yes

Reviewer #2: Yes

5. Is the manuscript presented in an intelligible fashion and written in standard English?

Reviewer #1: Yes

Reviewer #2: Yes

6. Review Comments to the Author

Reviewer #1: Perhaps a table with a summary of results could help to point out the main themes/categories/solutions based on the results

Reviewer #2: It was a pleasure reading this important work with fascinating findings. There are only a few items to address to strengthen the manuscript for publication.

(1) An expanded literature review;

(2) Counts/quantification of the focus group themes

(3) More discussion rooted in existing reward model structure

(4) Limitations of study improved

(1) The literature could include its own section to connect to existing work in this area. Journal of Research

Administration, Chronicle of Higher Ed, and others have research to give more context to scholarly communication, misconduct statistics, team science, and any additional context would help connect this work to the existing body of knowledge related to P&T.

(2) The direct quotes and themes are very good, but having counts of mentions or number of participants that represented/agreed with a theme or comment would give more detail to readers about how prevalent the sentiments were. Perhaps, even a table of themes and examples of each would present this in a more readable manner than the paragraphs of text.

(3) CRediT (Contributor Roles Taxonomy) may be an easy inclusion to explain how roles on teams do get credit now in some fields on large research teams. There are other metrics you can cite to connect the discussion and recommendations to current practices across the hard sciences. The entire discussion section would benefit from the authors adding more ideas and recommendations. What there is great, I just would like to see more.

(4) The limitations of the study are clear, but having a stand alone section of the limitations and ways to address them in future research would increase readability. Towards the end of that section a sentence just starts "Since we" and then is blank... so what should be there?

Overall, more subheadings and more concise writing style would also help.

7. PLOS authors have the option to publish the peer review history of their article (what does this mean?). If published, this will include your full peer review and any attached files.

Reviewer #1: No

Reviewer #2: **Yes: **Bradley Wade Bishop

---

## [Editor Report · Decision Letter 2]

12 Jul 2024

Title: Relational responsibilities: Researchers perspective on current and progressive assessment criteria: a focus group study

PONE-D-23-25545R2

Dear Dr. Tijdink,

We’re pleased to inform you that your manuscript has been judged scientifically suitable for publication and will be formally accepted for publication once it meets all outstanding technical requirements.

Kind regards,

Julian D. Cortes

Academic Editor

PLOS ONE
---

## [Editor Report · Acceptance letter]

24 Jul 2024

PONE-D-23-25545R2 

PLOS ONE

Dear Dr. Tijdink, 

I'm pleased to inform you that your manuscript has been deemed suitable for publication in PLOS ONE. Congratulations! Your manuscript is now being handed over to our production team.

Kind regards, 

on behalf of

Professor Julian D. Cortes 

Academic Editor

PLOS ONE